# Oxidative Stress Mediates Epigenetic Modifications and the Expression of miRNAs and Genes Related to Apoptosis in Diabetic Retinopathy Patients

**DOI:** 10.3390/jcm13010074

**Published:** 2023-12-22

**Authors:** Sarah Karam-Palos, Irene Andrés-Blasco, Cristina Campos-Borges, Vicente Zanón-Moreno, Alex Gallego-Martínez, Victor Alegre-Ituarte, Jose J. García-Medina, Salvador Pastor-Idoate, Inmaculada Sellés-Navarro, Jorge Vila-Arteaga, Antonio V. Lleó-Perez, Maria D. Pinazo-Durán

**Affiliations:** 1Ophthalmic Research Unit “Santiago Grisolía”/FISABIO, 46017 Valencia, Spain; sakapa@alumni.uv.es (S.K.-P.); irene.andres@fisabio.es (I.A.-B.); hecris@alumni.uv.es (C.C.-B.); vicaleitu@gmail.com (V.A.-I.); an.lleop@comv.es (A.V.L.-P.); 2Cellular and Molecular Ophthalmo-Biology Group, Department of Surgery, University of Valencia, 46010 Valencia, Spain; 3Department of Ophthalmology, University Hospital “Arnau de Vilanova”, 25196 Valencia, Spain; 4Net of Research in Inflammatory Diseases and Immunopathology of Organs and Systems “REI-RICORS” RD, Institute of Health Carlos III, 28029 Madrid, Spain; jj.garciamedina@um.es (J.J.G.-M.); spastor@saludcastillayleon.es (S.P.-I.); inmasell@um.es (I.S.-N.); 5Institute of Biotechnology, University of Porto, 4169-007 Porto, Portugal; 6Department of Preventive Medicine and Public Health, University of Valencia, 46010 Valencia, Spain; 7Department of Ophthalmology, University Hospital “Morales Meseguer”, 30008 Murcia, Spain; 8Department of Surgery, Pediatrics, Obstetrics and Ginecology, Faculty of Medicine, University of Murcia, 30100 Murcia, Spain; 9Institute of Applied Ophthalmobiology “IOBA”, University of Valladolid, 47002 Valladolid, Spain; 10Department of Ophthalmology, University Clinic Hospital of Valladolid, 47003 Valladolid, Spain; 11Department of Ophthalmology, University Hospital “Reina Sofia”, 30003 Murcia, Spain; 12Department of Ophthalmology, University and Polyclinic Hospital “La Fé”, 46026 Valencia, Spain; vila_jorart@gva.es; 13Innova Ocular Vila Clinic, 46004 Valencia, Spain

**Keywords:** diabetic retinopathy, oxidative stress, microRNAs, genes, apoptosis, epigenetics, neurodegeneration

## Abstract

Knowledge on the underlying mechanisms and molecular targets for managing the ocular complications of type 2 diabetes mellitus (T2DM) remains incomplete. Diabetic retinopathy (DR) is a major cause of irreversible visual disability worldwide. By using ophthalmological and molecular-genetic approaches, we gathered specific information to build a data network for deciphering the crosslink of oxidative stress (OS) and apoptosis (AP) processes, as well as to identify potential epigenetic modifications related to noncoding RNAs in the eyes of patients with T2DM. A total of 120 participants were recruited, being classified into two groups: individuals with T2MD (T2MDG, n = 67), divided into a group of individuals with (+DR, n = 49) and without (−DR, n = 18) DR, and a control group (CG, n = 53). Analyses of compiled data reflected significantly higher plasma levels of malondialdehyde (MDA), superoxide dismutase (SOD), and glutathione peroxidase (GPx) and significantly lower total antioxidant capacity (TAC) in the +DR patients compared with the −DR and the CG groups. Furthermore, the plasma caspase-3 (CAS3), highly involved in apoptosis (AP), showed significantly higher values in the +DR group than in the −DR patients. The microRNAs (miR) hsa-miR 10a-5p and hsa-miR 15b-5p, as well as the genes *BCL2L2* and *TP53* involved in these pathways, were identified in relation to DR clinical changes. Our data suggest an interaction between OS and the above players in DR pathogenesis. Furthermore, potential miRNA-regulated target genes were identified in relation to DR. In this concern, we may raise new diagnostic and therapeutic challenges that hold the potential to significantly improve managing the diabetic eye.

## 1. Introduction

Diabetic retinopathy (DR) is a microvascular–neurodegenerative complication of diabetes mellitus (DM) and the leading cause of visual loss in working-age people across many countries [1,2]. Sustained hyperglycemia, multi-metabolic pathway alteration, and a variety of endogenous/exogenous risk factors induce oxidative stress (OS) as well as preclinical changes in the retinal neurovascular unit (NVU) [3,4,5], as depicted in Figure 1. A longer duration of DM, along with high blood pressure (HBP), augmented body mass index (BMI), and dyslipidemia, significantly increases the risk of developing DR [3,4,6]. Outstanding strategies for DR management are emerging, including (1) innovative medical laser–surgical treatments, (2) chorio-retinal multimodal imaging, (3) electrophysiological probes of the retina and optic pathway, and (4) novel digital technology domains, including artificial intelligence (AI) and machine learning [3,6,7,8,9,10,11].

Imaging and molecular-genetic biomarkers are directed to identify DR risk factors, to detail its clinical signs, to evaluate its progression, and to classify the stage of retinopathy [8,9,10,11,12,13,14,15,16,17]. Epidemiological and experimental studies have reported an important interindividual variation in DR development and severity [18,19]. Thus, identification of more precise biomarkers that may help to stratify the risk, or to evaluate the therapeutic response of DR, is mandatory.

Excessive generation of reactive oxygen species (ROS) and the impossibility of counteracting their downstream effects with appropriate activity of the antioxidant defenses (such as enzymatic scavengers: superoxide dismutase (SOD) and glutathione peroxidase (GPx)) induce deleterious damage to the nucleic acids, lipids, and proteins [19,20]. In this OS atmosphere, ROS can amplify specific cell signaling proteins and transcription factors, such as mitogen-activated protein kinase (MAPK), nuclear factor (erythroid-derived 2)-like 2 (Nrf2), nuclear factor kappa B (NF-kB)), protein kinase C (PKC), p53, and others, leading to the activation of angiogenic, autophagic, inflammatory, and apoptotic pathways, with detrimental consequences [21,22].

Apoptosis (AP), programmed cell death, is essential during central nervous system (CNS) development. Exhaustive research has been performed in recent years for elucidating the apoptotic core machinery and the molecular-genetic pathways involved in this process [21,22,23,24]. In fact, cells can survive or die by apoptosis depending on the balance between proapoptotic and antiapoptotic signals within and around them. Figure 2 shows the intrinsic and extrinsic pathways of apoptosis. Each of these routes needs certain triggering signals to start an energy-dependent cascade of regulated molecular events. The proapoptotic mediators of the extrinsic (tumor necrosis factor (TNF)-related apoptosis-inducing ligand (TRAIL), Fas ligand (FasL), death-inducing signaling complex (DISC), Fas-associated death domain (FADD)-like IL-1β-converting enzyme-inhibitory protein (c-FLIP), TNFR1-associated death domain protein (TRADD), TNF-like weak inducer of apoptosis (TWEAK), and the nerve growth factor (NGF)) as well as the canonical intrinsic mitochondrial death (B-cell lymphoma 2 (Bcl-2) family proteins (group II and group III), proapoptotic homology (BH)3-only (including the following: Bcl2 modifying factor (Bmf), BIM, BID, Bcl-2 antagonist of cell death (BAD), p53 upregulated modulator of apoptosis (PUMA), and the phorbol-12-myristate-13-acetate-induced protein (NOXA)), Bcl-2-associated X protein (Bax), and the Bcl-2 homologous antagonist killer (Bak)) pathways, as well as the main antiapoptotic mediators from the intrinsic (Bcl-2 (group I), B-cell lymphoma-extra large (Bcl-XL)) and extrinsic (cellular FLICE (FADD-like IL-1β-converting enzyme)-inhibitory protein (c-FLIP) and nuclear factor kappa B (NF-κB)) pathways, finely orchestrate the balanced AP process and the switch between health and disease. It has been well established that AP is a key contribution to postembryonic development, to adult tissue homeostasis, and to the etiopathogenesis of multiple diseases, including cancer, neurodegenerative disorders, cardiovascular diseases, and ocular pathologies [21,22,25,26].

Small single-stranded noncoding microribonucleic acids (miRNAs/miRs) can regulate the expression of messenger RNAs and proteins within the cells and tissues in health and disease [27]. miRNAs commonly act by repressing gene expression and by binding to the 3’ UTR fragment, inducing mRNA degradation. It is important to remember that the miRNAs not only target genes, but also specific and significant sections of signaling pathways. In this context, changes in miRNA expression levels have been reported to be associated with the development and progression of DR [28], and these molecular actors have also been introduced as useful disease biomarkers. Furthermore, a crosslink between the expression levels of miRNAs and OS pathways has been widely investigated [4,29,30].

Epigenetics encompasses all changes involving gene activity that do not implicate alterations in DNA sequence, such as DNA methylation, modification of histones, modifications of chromatin structure, loss of imprinting, and noncoding RNA activities [31]. In this sense, the epigenome entails these genome modifications that do not affect the DNA sequence, selectively choosing, under specific conditions, whether genes will be switched “on” or “off” [32]. It has been described that miRNAs contribute to a wide spectrum of disorders by means of epigenetic regulation of gene expression [32,33,34]. In fact, epigenetic modifications are remarkably involved in T2DM [35]. However, the human epigenome (in great proportion) remains unknown, creating an increasing interest in this regard in relation to DR research.

Because of this, evaluating the OS-related pathways, as well as the expression levels and biological functions of target miRNAs, is capital to understanding DR ethiopathogenesis. Therefore, we have taken into consideration that supramolecular targeted diagnosis, encompassing epigenetic modifications and signal transduction pathways, may play a more promising role in managing diabetic eyes.

Therefore, the main purpose of this work is to report on the current level of knowledge on the clinical imaging and molecular-genetic landmarks that may help enlighten us to T2DM risk factors and pathogenic mechanisms to better understand the disease, and to elucidate potential applications for eye care in diabetics. By using ophthalmological imaging approaches (ocular biomicroscopy (BMC), ocular fundus examination (OF), retinography (RTG), optical coherence tomography (OCT)) and experimental tests (enzyme-linked immunosorbent assay (ELISA), next-generation sequencing (NGS), and real-time polymerase chain reaction (RT-PCR)), we registered a series of information to build a data network that may help in addressing new diagnostic challenges for DR in T2DM patients.

## 2. Methods

### 2.1. Study Design and Participant Characteristics

The present study obtained Institutional Board approval (ref: CEIm 12/2022 and 42/2022). The procedures used adhere to the tenets of the Declaration of Helsinki (first version 1964; last review, led by the American Medical Association (AMA), currently in progress) involving medical research with human subjects, the European Commission rules for research, and the ARVO guidelines regarding the ethical use of human subjects in research.

This prospective cohort study was designed to compare demographic, clinical, biochemical, and molecular-genetic data obtained from all volunteers, which were anonymized according to current legislation to preserve human rights regarding medical data.

Demographic information and characteristics were registered in a data sheet in the Microsoft Excel program as DEMO, including code, age, sex, personal and familial characteristics, lifestyle, main comorbidities, and therapies.

Ophthalmological examination was initially carried out on 162 eyes, which were registered in a data sheet in the Microsoft Excel program as OPHTHAL.

Finally, participants were classified into two main groups, attending to the inclusion/exclusion criteria depicted in Table 1, by using clinical measurements and grading of ophthalmic images, as shown in Figure 3: T2DM without DR (T2DM−DR), T2DM with diabetic retinopathy (T2DM+DR), and control individuals (CG).

Initially, 162 volunteers were recruited at the ophthalmological clinics, according to the inclusion/exclusion criteria. A total of 120 Caucasian individuals of both sexes aged 40-80 years were finally considered for this study and distributed into DR patients and CG subjects. Cases and controls were classified according to the inclusion and exclusion criteria (see Table 1 and Figure 3).

The DR group was identified, by both clinical history data and ophthalmologic examination, by retina specialists from the collaborative hospitals pertaining to the Spanish Public Health Service (Arnau de Vilanova—Valencia, Clinic—Valladolid, “Reina Sofia”—Murcia, “Morales Meseguer“—Murcia, and Polytechnic and University “La Fe”—Valencia). Data were collected from both eyes regardless of the affectation, but only the right eye (RE) was considered for statistical processing and final data. A total of 67 participants constituted the T2DM group (see Figure 3).

Control subjects presented a normal anterior and posterior eye segment at biomicroscopic examination. The RE was only considered for this study. A total of 53 participants constituted the CG of the study (see Figure 3).

Each ophthalmologist reported anonymously the corresponding data from each study volunteer by means of a self-designed report form that was entered and processed using a Microsoft Excel spreadsheet for Windows. Figure 3 shows the participant distributions in each group and subgroup of this cohort.

### 2.2. Proceedings

The present work was a cross-sectional, observational, and correlational case-control study for evaluating the risk factors, lifestyle, clinical data, and biochemical and molecular-genetic facts that might help to identify diabetics at risk of developing DR and/or progressing through the ocular disease and vision loss.

Through the following subsections, we describe the participants’ characteristics and lifestyle, ophthalmological examination proceedings, sampling protocols, and the experimental assays used, as well as the information obtained from these assays and the corresponding statistical analyses that were conducted in this study.

#### 2.2.1. Demographic Characteristics of the Study Participants

Demographic data and participant characteristics were registered for each subject. The registered variables were age, gender, family history of DM, DR, HBP, and body mass index (BMI). Also, physical exercise adherence as well as drinking or smoking habits were recorded.

#### 2.2.2. Clinical Characteristics of the Study Participants

Diagnosis of T2DM was defined when at least one random blood glucose value was higher than 200 mg/dL, fasting blood glucose was higher than 126 mg/dL, the ciphers of HbA1c were higher than 6.5% (48 mmol/mol), the above data were retrieved from medical records, or the self-reported clinical history of the disease was checked. Individuals with a T1DM diagnosis were considered to fit the absolute exclusion criteria. Control individuals were also interviewed for signs or symptoms of any disorder, current treatment, recent surgery, and family history of DM, DR, and HBP.

Duration of T2DM was considered as the lead time between the diagnosis of the disease and the first review of the medical records (clinical history) of the patient, and the corresponding data were recorded. Current therapy for DM as well as previous history of DR/DME laser treatment and/or surgery and the type of proceeding were also recorded.

Ophthalmological examination included best corrected visual acuity (BCVA) in LogMAR, IOP measurement, slit-lamp biomicroscopy of the anterior eye segment and media, dilated stereoscopic fundus examination, and macular analysis using the Cirrus Spectral domain OCT 5000 (Carl Zeiss Meditec, Inc., Madrid, Spain) with special interest in detecting the retinal nerve fiber layer (RNFL) characteristics. The reference for the software of this tool is 11.0.0.29946.

#### 2.2.3. Sample Proceedings

Biological samples were collected from each participant. Blood and tears were alternatively collected from the participants, according to their respective processing protocols.

Blood was collected via venipuncture of the antecubital vein under fasting conditions (10–11 h). Vacutainer^®^ (Becton, Dickinson and Co., Franklin Lakes, NJ, USA) blood collection tubes were used and immediately labeled. Plasma was obtained via centrifugation of whole blood samples, aliquoted, labeled, and stored at −80 °C until processing.

Tear samples were collected by gently rubbing the inferior meniscus of both eyelids with a microglass pipette. Reflex tears (20–30 μL) were deposited in a 0.2 mL Eppendorf, labeled, and stored at −80 °C until processing, as previously reported [36,37].

The biochemical, molecular, and genetic information from each participant was expressed as the mean ± SD or percentages for 2–3 experiments for each participant assay, and the results were registered in a data sheet in the Microsoft Excel program, specially designed for this study, as LAB.

#### 2.2.4. Biochemical and Molecular-Genetic Variables of the Study Participants

##### Clinical Biochemistry

The biochemical analyses were carried out on blood samples. Routine biochemistry determinations (basal glycemia, glycosylated hemoglobin (HbA1c), total cholesterol (T Chol), high-density lipoprotein (HDL) cholesterol, low-density lipoprotein (LDL) cholesterol, very low-density lipoprotein (VLDL) cholesterol, triglycerides (TRIG), urea, creatinine (CREAT), and iron) were performed using automated chemistry analyzers at the Clinical Analysis Department of the main study center in Valencia (Spain), as follows: (1) Abbott kits manufactured for being used with Architect c8000 (Abbott Laboratories; Abbott Park, IL, USA) and (2) Arkray AU 4050 (Arkray Global Business Inc., Kyoto, Japan). All procedures were supervised by a specialist in clinical analyses.

##### Oxidative Stress

Malondialdehyde (MDA) was determined with the MDA-thiobarbituric acid reactive substance (TBARS) assay. Briefly, plasma samples from DR patients and the CG were treated with HCl and SDS. The MDA present in samples reacts with TBA and the fluorescent complex formed was measured in a Fluoroskan^®^ Ascent FL (Thermo Electron Corporation, Philadelphia, PA, USA), as reported elsewhere [38,39,40].

Superoxide dismutase was determined in plasma samples using the Superoxide Dismutase (Ransod) assay (Randox Labs, Barcelona, Spain). We followed the manufacturer’s instructions for performing the assay, as in our previous work [41].

Glutathione peroxidase (GPx) was measured in plasma using the Glutathione Peroxidase Assay Kit (Cayman Chemical Co., Ann Arbor, MI, USA), following the manufacturer’s instructions and according to our previous assays [41,42].

Total antioxidant capacity (TAC) was analyzed in plasma samples by means of the Total Antioxidant Assay Kit (Cayman Chemical Co., Ann Arbor, MI, USA), according to the manufacturer’s instructions [39,43].

##### Apoptosis

Caspase-3 (CAS3) concentration in plasma samples was determined via ELISA, by using a commercial kit (human CAS3 (active) ELISA kit: ref. KHO1091; Invitrogen, Vienna, Austria). Optimal dilutions were determined, and the assay was performed according to the manufacturer’s instructions as described elsewhere [44,45].

##### MicroRNAs

The miRNA fingerprints were assessed in tear samples from each group of participants. Tear samples were conveniently defrosted and prepared for RNA extraction by using the miRNeasy Mini Kit (QIAGEN Inc., Hilden, Germany). The purification of the preparation was performed by means of spin column chromatography, with an appropriate resin as the separation matrix. Next, the quality/quantity of total tear RNA was obtained with both a Bioanalyzer 2100 (Agilent^®^ Technologies, Inc., Santa Clara, CA, USA) and the RNA 6000 Nano Kit (Agilent^®^ Technologies, Inc.). Then, the RNA libraries were prepared by using the NEBNext^®^ Multiplex Small RNA Library Prep Set for Illumina^®^ (#E7300 y #7580; New England BioLabs^®^, Inc., Ipswich, MA, USA), following the manufacturer’s instructions. Following the guidelines for low-RNA-concentration samples, as in the case of the tear samples, the adapters and RT primers were diluted 1:2 with nuclease-free water, and 15 cycles were utilized during the amplification via PCR proceedings. The purification of indexed libraries was carried out using the QIAquick^®^ PCR Purification Kit (#28104, QIAGEN^®^, Hilden, Germany). Quality control of the obtained libraries was performed with a 4200 TapeStation (Agilent^®^ Technologies, Inc.) and a High-Sensitivity D1000 Kit (Agilent^®^ Technologies, Inc.). Finally, the miRNA fraction of each library (120–200 bp) was collected with the Pippin Prep System (Sage Science, Inc., Beverly, MA, USA) using 3% agarose dye-free gel cassettes with internal standards (Marker P) (Sage Science #CDP3010). Quantification of miRNAs was performed by using a 4200 TapeStation (Agilent^®^ Technologies, Inc.) and a High-Sensitivity D1000 Kit (Agilent^®^ Technologies, Inc.) prior to normalization and pooling. Finally, sequencing was performed on a NextSeq 500 System (Illumina, Inc., San Diego, CA, USA) with a mid-output flow cell for 150-cycle reads, which permitted us to obtain ≈ 3.5 million reads per tear sample [46,47]. FASTQ file quality was assessed using the FASTQC tool (https://www.bioinformatics.babraham.ac.uk/projects/fastqc/, accessed on 10 September 2023). Adapters and low-liability reads were removed. The noncoding RNAs previously described in the ENSEMBL database were identified and characterized. The new PmiRtarbase [48] was used for searching and downloading all related entries for the miRNAs. Bioinformatics was performed using the Limma and edgeR packages deposited in Bioconductor (www.bioconductor.org, accessed on 15 September 2023).

##### Gene Expression

Whole blood samples collected from the study participants were placed into 4.5 mL EDTA tubes. Total RNA was isolated from each blood sample by using the TRIzol method [49]. Next, an amount of 300 ng of total RNA (integrity number: RIN > 7) was converted into cDNA via reverse transcription using the High-Capacity RNA-to-cDNA™ Kit (Applied Biosystems, Foster City, CA, USA). The relative *BCL2L2* and *TP53* gene expression was analyzed via qRT-PCR using a 7900HT Sequence Detection System (Applied Biosystems^®^, Madrid, Spain). Then, the TaqMan gene expression protocols were followed for targeting (*BCL2L2*, *TP53*) and internal control (*18S rRNA*) genes (Applied Biosystems^®^, Spain) [50,51]. Each blood sample was assayed in duplicate. Gene expression values were estimated using the specific formula “double delta Ct”, as previously shown [50,51]. All data were expressed as fold changes in gene expression for each group and subgroup of the study participants.

#### 2.2.5. General Statistics and Bioinformatics

All the described experiments were performed in duplicate for each sample. Quantitative variables were summarized using the mean ± standard deviation (SD). Categorical variables were expressed as percentages, and statistical differences were measured using the chi-squared test. The Shapiro–Wilk test was used to assess normality in each variable. Student’s *t*-test was used to compare normally distributed variables among controls and glaucoma patients, whilst for non-normally distributed variables the Mann–Whitney U test was used. For multiple comparisons, an ANOVA test was conducted. Differences in qualitative variables between the two groups were compared using Fisher’s test. The measurement data are expressed as the mean ± SD. Statistical analyses were performed after the outliers were removed. The sensitivity and specificity of each independent variable were obtained for a range of different cut-off points. The level of statistical significance was set at *p* < 0.05. All statistical analyses were performed using R Statistics v 4.0.0 (R Foundation for Statistical Computing, Vienna, Austria). The sample size of this study provided >80% power to detect the risk of progressing DR among the T2DM cases, as compared with the CG, at a two-sided alpha level of 0.05. Two-sided tests at the 0.05 level of significance were used for all statistical comparisons.

## 3. Results

Accurate information for the main purpose of this study, raising DR awareness, was collected from our data sets for 120 participants.

The mean age of participants did not show statistical differences between groups and subgroups (*p* = 0.215 (ANOVA)), and it was 52 ± 10 years for the CG, 55 ± 11 years for the T2DM−DR patients, and 56 ± 12 years for the T2DM+DR patients.

Distribution by gender did not display statistical significance between the study participants, and it was 13% men/39% women for the CG, 16% men/12% women for the T2DM−DR group, and 19% men/21% women for the T2DM+DR group.

In subsequent sections, we depict the above collected data and the results of their analyses.

### 3.1. Demographics and Participant Characteristics

After an initial recruitment of a potential 162 volunteers of both sexes (75 T2DM patients and 87 healthy controls), our study population was composed of 120 participants, which were classified into 52 participants (43.3%) in the CG, 28 diabetics (23.3%) constituting the T2DM−DR group, and 40 diabetics (33.3%) forming the T2DM+DR group, as shown in Figure 3. Tobacco and alcohol habits, as well as the BMI and physical activity information from each participant, were recorded, and the results showed significant differences between groups related to these variables, especially regarding the presence of HBP and T2DM (and family history) in the study participants, as well as physical activity (Table 2).

### 3.2. Ophthalmologic Evaluation

Based on the inclusion/exclusion criteria for participating in this study, data collected from the anterior and posterior eye segment examinations of the two T2DM subgroups (T2DM−DR, T2DM+DR) and the healthy controls (CG) were analyzed, and the results are shown in Table 3.

The BCVA, expressed as the logarithm of the minimal angle of resolution (LogMAR), was significantly lower in the T2DM+DR patients as compared with the T2DM−DR subgroup and the CG (*p* = 0.014).

The IOP values were quite similar between the study participants, and the results lacked statistical significance (*p* = 0.912). Also, the distribution of frequencies of the variables pertaining to the anterior eye segment examination did not show statistical differences between groups and subgroups (*p* = 0.055, χ^2^ de Pearson).

Evaluation of the ocular fundus (OF) and the qualitative/quantitative macular examination with the OCT were normal in the T2DM−DR group and the CG of participants. However, the central subfield thickness (CST) and the cube volume (CV) showed noticeably higher values in the T2DM+DR subgroup than in either the T2DM−DR patients or the CG (Table 3).

### 3.3. Biochemical Variables

Comparison of the most common clinical biochemical variables between groups and subgroups of participants is reflected in Table 4. In fact, the main variables from the clinical biochemistry analyses carried out on our study participants (basal glycemia, HbA1c, T Chol, HDL, LDL, VLDL, and TRIG) showed a higher differential expression profile in the T2DM subgroups than in the CG, except for urea, creatinine, and iron, which lacked statistical significance.

We also performed bivariate correlation analyses between the OCT parameters and the biochemical and lifestyle characteristics, and our data showed a significant lineal relation between the macular CST and CV with the biochemical values of the HbA1c and the lipidic profile (Total Chol, LDL, and VLDL). Table 5 summarizes the analyzed correlations with the Pearson correlation coefficient (significance levels: 0.05 and 0.01). Furthermore, physical exercise habits showed significant differences with the macular CV (*p* = 0.003; ANOVA). The rest of the parameters lacked statistical significance.

### 3.4. Molecular-Genetic Variables

Overall data processing gave us a set of molecules presumptively involved in DR pathogenesis. Qualitative and quantitative characteristics of each molecule and/or gene are considered below.

#### 3.4.1. Oxidative Stress

Over time, OS leads to structural/functional eye damage. Comparison of data obtained from plasma samples of the study participants (Figure 4) revealed significantly higher OS activity in the diabetic groups versus the controls. Furthermore, it was observed that there was significantly elevated OS activity in the T2DM+DR patients versus the CG, as reflected by the MDA/TBARS (*p* = 5.97 × 10^−24^), SOD (*p*= 3.62 × 10^−16^), and GPx (*p* = 0.025) expression levels (Figure 4A–C, respectively). Significantly lower plasma TAC values from the T2DM+DR patients versus the CG (*p* = 8.80 × 10^−8^) (Figure 4D) were observed.

#### 3.4.2. Apoptosis

CAS3 plasma levels were assayed in the study participants, and the data from the CG were 0.05 ± 0.02 ng/mL. However, in the T2DM+DR group these values were 0.12 ± 0.05 ng/mL, and in the T2DM−DR group they were 0.08 ± 0.03 ng/mL.

Hence, the plasma differential expression profile of the above AP marker was statistically significant in the T2DM+DR group with respect to the T2DM−DR group (*p* < 0.001) and the CG (*p* < 0.0001).

#### 3.4.3. miRNAs

Statistically significant differential miRNA expression fingerprints were seen in tear samples of the study participants.

In fact, of the 62 miRNAs identified in tears, 16 miRNAs showed different expression profiles in diabetics versus the CG, with the majority involved in OS, AP, cell cycle regulation, angiogenesis, and inflammation pathogenic processes.

Table 6 illustrates the miRNAs that appeared significantly upregulated or downregulated in tear samples from the study groups and subgroups, based on the RNAseq results. The groups with superscript ^(1)^ correspond to those that were used as the reference group in each comparison. Overall, our data showed that miR-10a-5p and miR-15b-5p were downregulated and upregulated (respectively) in the T2DM+DR group with the highest statistical significance (highlighted in black).

Figure 5 shows the miR-15b-5p and miR-10a-5p differential tear signatures in the groups and subgroups of participants.

After qRT-PCR analysis, tear miR-15b-5p expression levels (Figure 5C) displayed noticeable upregulation in both the T2DM−DR and the T2DM+DR subgroups when compared with the CG (*p* = 0.199, Kruskal–Wallis test). Furthermore, we observed a slight decreasing trend in tears from the T2DM+DR, as compared with the T2DM−DR (*p* = 0.401, pairwise comparison). The miR-15b-5p inhibits the expression of the target genes, such as the antiapoptotic *TP53* gene, and, in turn, may alter the AP process in diabetics.

Tear miR-10a-5p expression levels (Figure 5D) showed statistically significant differences (*p* = 0.021) when the three groups were analyzed together (T2DM−DR vs. T2DM+DR vs. CG; Kruskal–Wallis test). After pairwise comparisons, a statistically significant downregulation of this miRNA was observed in tears from the T2DM−DR patients as compared with the CG (*p* = 0.019). Likewise, a statistically significant downregulation was observed in tears from the T2DM+DR patients when compared with their corresponding healthy counterparts (*p* = 0.009). The expression of this miRNA was noticeably reduced in the T2DM+DR group with respect to the T2DM−DR group, although the difference did not reach statistical significance (*p* = 0.227).

In this context, the lower levels of this miRNA lead to a decreased function on their target genes, such as the proapoptotic *BCL2L2L* gene (score 87); the score indicates the high-specificity miRNA-target gene, which is high with respect to miR-10a-5p and this gene.

#### 3.4.4. Gene Expression

The following predicted target genes (analyzed in blood samples) of the above miRNAs showed statistically significant differential fingerprint in T2DM+DR patients.

The *BCL2L2* gene (B-cell lymphoma): Bcl-2-like protein 2 is a 193-amino acid protein that in humans is encoded by the *BCL2L2* gene on chromosome 14 (band q11.2-q12) and highly implicated in AP. Bcl-2 is a regulatory factor of cell death, located in the outer mitochondrial membrane (see Figure 1), with the role of inhibiting the function of proapoptotic genes. Our data obtained from blood samples of the study participants are reflected in Figure 6A,C.

The *TP53* gene (tumor protein 53 gene), located in the short arm of chromosome 17 (17p13.1), encodes nuclear proteins that bind to DNA and regulate gene expression to prevent mutations of the genome. p53 plays an important role in regulation/progression through the cell cycle, apoptosis, and genomic stability. The *TP53* gene is a tumor suppressor, also known as the genome guardian. The *TP53* gene expression in blood samples from this cohort is shown in Figure 6B,D.

Therefore, based on the clinical, retinal imaging, biochemical, and molecular-genetic approaches and their crosstalk processes, we propose a new open window to personalized potential DR diagnosis, to better manage diabetic eyes and vision.

## 4. Discussion

We intended to shed light on the crosslink between OS and AP, as well as what epigenetic modifications may occur in DR patients. By collecting data from risk factors, clinical facts, retinal imaging, and pathogenic mechanisms, we built a data network that allowed us to obtain interesting results. We found that chronic hyperglycemia is responsible for OS status and its downstream effectors, which, in turn, induces a series of changes in signaling pathways, especially affecting the AP process. The OS biomarkers (MDA, SOD, GPx, TAC), the AP biomarkers (CAS3), the miRNAS 10a-5p and 15a-5b, and their predicted target genes *BCL2L2* and *TP53* have been identified as important players in DR in our study population. Characterizing these molecular actors and improving our knowledge about the crosstalk between them could open up some innovating paradigms with the main purpose of improving eye and vision care in diabetics.

Demographic and ophthalmologic characteristics of our cohort permitted us to detect that poor glycemic control, long-lasting T2DM, overweight/obesity, the presence of HBP and dyslipidemia, and familial history of DM were associated with a higher risk of DR and DR progression, confirming the growing body of evidence in recent reports [1,2,3,10,11,12,13,14,52].

We performed bivariate correlation analyses to achieve greater knowledge of our clinical, retinal imaging, and biochemical analyses of our study participants, and our results reflected a significant lineal relation between the CST and CV (obtained via macular OCT examination) with the HbA1c values and the lipidic profile (Total Chol, LDL Chol, and VLDL Chol). Bjornstad et al. [53] described the relationship between the retinal thickness and morphological changes on OCT from T2DM patients in the TODAY study, regarding the association between elevated HbA1c and increases in total retinal thickness. Also, Liang et al. [54] reported that in T2DM patients with DR, the T Chol, LDL Chol, and HbA1c were closely correlated with CST. In a similar manner to us, Sasaki et al. [55] described in a T2DM cohort that higher LDL Chol levels were associated with increased CST and CV in diabetics with mild DR without macular edema.

OS is known as the process in which the imbalance between the pro-oxidants and antioxidants (with higher proportion of pro-oxidative sources) damages cells, tissues, and organs. At a physiological level, ROS participate in the regulation of lysosomes and mitochondria. The mitochondria form ATP via the electron transport chain and oxidative phosphorylation, resulting in ROS production. However, a higher ROS level is harmful for cells and tissues, and the subsequent mitochondrial dysfunction severely damages the lysosomes. Therefore, under pathological conditions these systems fail and generate large amounts of pro-oxidants [4,13,14,18,19,28,38,43,46,56,57,58,59]. Hyperglycemia leads over time to OS, which induces structural and functional damage to the retinal cell phenotypes, creating a harmful increased pro-oxidant and decreased antioxidant atmosphere [4,18,19,28,60,61,62,63].

In the present work, we found significantly higher levels of plasma MDA/TBARS and significantly lower plasma levels of SOD, GPx, and the TAC in T2DM vs. the CG, as is indicative of OS status. Other authors also reported similar findings to ours. Marsen-Bouterse et al. [64], Domanico et al. [65], Pan et al. [66], Khan et al. [67], and Wu et al. [68], among many other authors worldwide [4,18,19,28,59,60,61,62,63,69], identified OS as a milestone in the onset and progression of DR. From a functional viewpoint, when cells are under stress, ROS can induce cell death via the extrinsic or intrinsic pathways (see Figure 2). In this context, pathologic ROS overproduction induces AP cell death in the diabetic retina, as previously reported [24,25,26,69]. Regarding this, Xie et al. explored the role of mtDNA oxidative damage in high-glucose-induced dysfunction in human retinal vascular endothelial cells [70].

Some relevant pathways induce upregulation of proapoptotic gene expression, and downregulation of neuroprotective factors. NF-κB is an important redox-sensitive transcription factor whose activation initiates a proapoptotic program. It has been recently reported that the signaling pathways of important transcription factors involved in OS, such as Nrf2, and inflammation, such as NF-κB, regulate the physiological redox status in the cells, subsequently modulating the cellular response to the above pathogenic mechanisms [71]. As described herein, we identified specific AP molecules in biological samples from T2DM patients and healthy CG individuals. First, we looked for the expression of plasma CASP3, which is a cysteine–aspartyl protease encoded by the *CASP3* gene that crucially mediates the activation cascade of CAS, which, in turn, is responsible for programmed cell death (see Figure 2). At the very onset of AP, CAS3 proteolytically cleaves the poly (ADP-ribose) polymerase (PARP) and initiates the sequence of facts leading to cell death. The most important PARP-1 function is DNA repair in response to a wide variety of exogenous/endogenous cellular stresses. In response to DNA damage, the activation of PARP-1 is an outstanding mechanism to keep cell homeostasis or to induce AP. The cleavage of PARP-1 by the above CAS leads to the generation of two fragments: (1) the 85-kD catalytic fragment and (2) the 24-kD DNA-binding domain. In this context, when comparing our results for the plasma samples from the T2DM+DR patients with those from their counterparts, significantly higher CAS3 expression was observed. The intrinsic apoptotic pathway is characterized by mitochondrial dysfunction and CAS activation (see Figure 2). Jänicke et al. [72] reported that CAS3 plays a pivotal role in the amplification of AP signaling via direct/indirect stimulation of the CAS downstream. CAS3 activation is a hallmark of AP and has been rendered a point of no return in the AP signaling molecular series. The AP of retinal cells has been pointed out as a preclinical sign of neurodegeneration in DR, as reported by Valverde et al. [73]. These authors studied the neurodegeneration phenomenon that occurs early in the diabetic retina, concluding that disbalance between the intracellular signaling proapoptotic and survival factors was linked to the severity of damage in the retinal cell phenotypes in the course of DR.

Based on the OS and AP molecular data obtained in this population study, one can hypothesize that over time hyperglycemia induced the formation and accumulation of ROS (superoxide anion (O2•−), hydrogen peroxide (H_2_O_2_), and hydroxyl radical (·OH)) that was reflected in a significant plasma MDA elevation and decreased SOD, GPx, and TAC in T2DM+DR patients. In addition, the increased OS core spread the oxidative damage which triggered the expression of transcription factors and target genes involved in AP, as exposed below.

We are trying to improve knowledge on the role of epigenetics in DR. As previously stated, epigenetics includes changes related to gene activity that do not involve DNA sequence alterations, such as DNA methylation, modification of histones, and noncoding RNAs, with the latter being important regulatory molecules (long-distance connectors and reprogramming factors) [31,32,33,34,35]. In fact, several genes associated with oxidative stress in DR pathogenesis are influenced by specific epigenetic modifications [74]. Maghbooli et al. [75] studied 162 T2DM patients, and a relationship between genetic and epigenetic associations with the onset of DR was described. In spite of this, up to today the human epigenome (in great proportion) remains elusive. We studied the tear fluid expression levels and biological functions of miRNAs and target genes in DR ethiopathogenesis. Hence, the tear expression levels of miRNAs in T2DM patients +DR and −DR were assayed and compared with those from the CG. From all miRNAs (62) identified in tears from the study cohort (see Table 6), 16 displayed differential signatures in diabetics versus the CG, with important contributions to several pathophysiological processes, such as OS, AP, cell cycle regulation, angiogenesis, and inflammation. We found that both miR-10a-5b and 15b-5p displayed differential signatures in the groups and subgroups of participants. These two miRNAs are highly involved in AP [76]. The human gene that encodes miR-10a-5p is located upstream of the *HOXB4* gene on chromosome 17q21. The 5p means that the miRNA is from the 5’ arm of the hairpin. miR-10a-5p has been implicated in p53/stress response pathway regulation via its repression of a wide number of the key genes of the p53 network [77]. In this sense, we have shown that miR-10a-5p was significantly downregulated in tears from the T2DM+DR patients, as compared with those from the CG. On the other hand, miR-15b-5p is encoded by the *MIR15B* gene, located on 3q25.33, and widely involved in the pathogenic mechanisms of several conditions, such as Alzheimer’s and Parkinson’s diseases, cerebral stroke, and diabetic complications, as well as having a dual role (oncogenic and tumor suppressor) in cancer [78]. In this work, we observed a noticeable upregulation of miR-15b-5p in the T2DM−DR and T2DM+DR subgroups when compared with the CG. Our findings characterize 10a-5p and 15b-5p miRNAs and their fingerprints in the regulatory network of DR in our T2DM patients. As ROS play relevant roles in a wide spectrum of intracellular signaling pathways by acting as second messengers, it may be hypothesized that in the presence of OS these miRNAs may intervene by targeting some nuclear proteins that control cell cycle, DNA repair, and AP, thus inhibiting the proliferation of the endothelial cells and inducing AP.

Because accumulating reports highlight pivotal roles for specific miRNAs in a variety of diabetic complications, we deal with studying the AP miR/target gene axis in T2DM patients with and without DR. In our study cohort, we saw that the *BCL2L2* gene and the *TP53* gene were upregulated in blood from T2DM+DR patients with respect to the CG (see Figure 6) [79,80]. Therefore, our bioinformatic proceedings predicted that miR-10a-5p downregulation in tears was correlated with a decreased expression of their target genes, such as the *BCL2L2L* and *TP53* genes, with regulatory AP functions. Also, the tear expression of miR-15b-5p was noticeably upregulated in the T2DM+DR patients, inhibiting the expression of their target genes, such as the *TP53* gene, thus blocking the AP process in diabetics. Therefore, differences in gene expression observed in this cohort can be understood as OS-related epigenetic modifications, regulated by a specific miRNA signature. We may propose that the miRNAs identified herein, and their target genes/signaling pathways, will help to achieve a better understanding of DR’s molecular and genetic bases.

Due to the enormous socioeconomic impact of DR [81], it has to be considered that personalized medicine is a challenge for managing diabetic eyes and vision, being underpinned by larger scientific knowledge on the cellular and molecular-genetic basis of DM and its ocular complications. Our main goal is to improve current information about DR. In our cohort, the described clinical, molecular-genetic candidate biomarkers may potentially lead to ameliorating the diagnosis and to developing new therapies to improve eye and vision care in diabetics.

In an attempt to improve the prognosis of T2DM patients with and without DR, three potential therapeutic/prognostic approaches can be suggested: (1) avoiding or mitigating OS, (2) enhancing the cell survival pathways, and (3) blocking the AP cascade via CAS inhibitors and other biological strategies.

The present work has a series of strengths. DR diagnosis was performed by an ophthalmology specialist on the retina sections, which reinforces validity, and the collection of clinical and imaging data was performed via directly using the medical history, personal interview, and ophthalmological examination of each participant. Statistical processing was performed for the RE of each participant, taking into consideration that only initial NPDR cases were recruited for the present study. To our knowledge, miR-10a-5p, miR-15a-5b, and their target genes *B2CL2L* and *TP53* were identified for the first time in relation to DR pathogenesis in the present work.

In contrast, some study limitations have also been considered by the research team. A relatively small sample size when classifying suitable study participants is present. Underestimation of DR cases is beyond our control. Some missing data were unrecoverable from the clinical histories and the personal interviews. It has to be considered when interpreting the results that an overstatement of the statistical power can appear in some data. Selection of candidate reference genes was performed via an extensive search of the literature and genes. RNA-seq-based assays will permit us to select and validate more appropriate candidates for future studies. Finally, we collected tears to evaluate the possibility of identifying miRNAs and their differential fingerprints in diabetic eyes (this is demonstrated herein). Due to the small number of tear samples, it was not possible to assay the predicted target genes, which were analyzed in blood samples. In the foreseeable future, we are planning to perform both assays on the two samples from each participant. Finally, we would like to state that the results of the qRT-PCR analysis of miR-15b-5p expression showed a noticeable trend toward upregulation in the diabetic groups, as compared with the CG. In addition, we found via qRT-PCR a slight downregulation of this miRNA in tears from the T2DM+DR when compared with the T2DM−DR.

In our opinion, pathological changes occurring in the choroid, retina, and vitreous can be reflected in the whole eye, including the ocular surface components, either via simple diffusion through the sclera or cornea, or via local and systemic circulation, and vice versa. Nevertheless, by analyzing tear film samples instead of the aqueous humor, vitreous, and/or blood for target molecules and gene concentrations, we can manage to create a very attractive promising window for the diagnosis of retinal diseases, as in the case of DR.

We suggest that supramolecular targeted diagnosis, as depicted herein, may play an important role in DR management. In this atmosphere, we propose that chronic hyperglycemia leads to OS and alteration of the retinal NVU, which in the presence of concrete risk factors induces relevant changes in a wide variety of molecules that we have identified in tears and plasma samples, such as MDA, SOD, GPx, CAS3, miR-10a-5p, miR-15b-5p, and the *BCL2L2* and *TP53* genes.

## Figures and Tables

**Figure 1 jcm-13-00074-f001:**
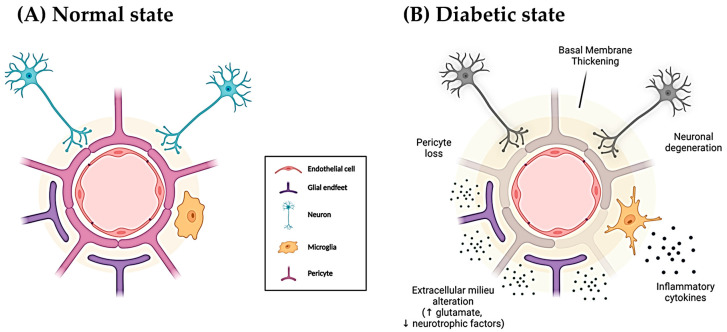
**Schematic drawing of the normal (A) and pathologic (B) retinal NVU.** Under normal conditions (**A**), the endothelial cells that form the retinal blood vessels are surrounded by three main cellular structures: glial end-feet, neuron dendrites, and pericytes. All these elements alongside the microglia act in a synergic way, regulating retinal tissue’s homeostasis according to its physiological requirements, guaranteeing BRB integrity and functionality. The diabetic condition (**B**) and its characteristic and maintained hyperglycemia cause an alteration in the mentioned elements’ functionality and viability. In this sense, pericytes and neurons degenerate, microglia activate in response to the harmful stimulus of diabetes, and the extracellular milieu is altered, leading to a thickening of the basal membrane and ultimately to a blood–retinal barrier dysfunction. NVU: neurovascular unit. ↓: decrease. ↑: increase. Figure created by A.G.-M. with BioRender.com, accessed on 10 October 2023.

**Figure 2 jcm-13-00074-f002:**
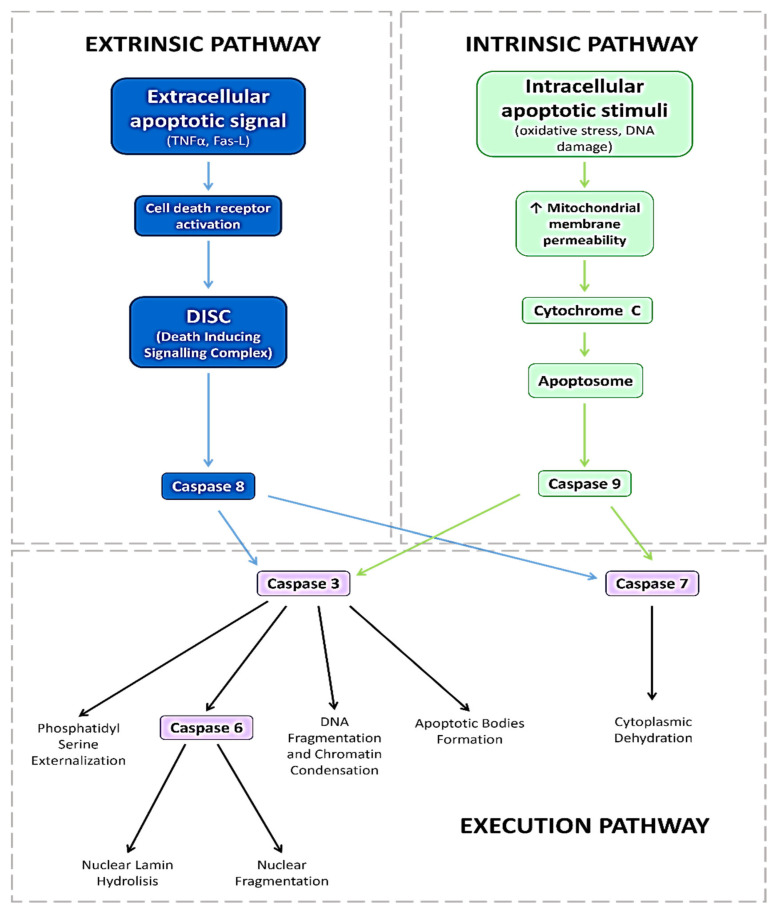
**Simplified schematic representation of the two AP pathways (intrinsic and extrinsic), the corresponding players, and their main biological functions.** Two converging but distinct pathways start AP: the extrinsic pathway (also known as death receptor AP pathway), and the intrinsic pathway (also named the mitochondrial or BCL-2-regulated AP pathway). The extrinsic death receptor pathway begins with the recognition of some extracellular apoptotic signals, such as certain members of the TNF family and Fas-L, by their specific cell membrane receptors. The binding between the receptor and its specific molecule leads to its activation and DISC recruitment, which activates caspase-8, which in turn activates the effector CAS3. In the intrinsic mitochondrial pathway, AP is initiated in response to diverse stimuli, including OS, that pathologically alter the mitochondria. This pathway is highly regulated by the BCL-2 protein family members (pro-AP: BIM, PUMA, BID, BMF, NOXA, BIK, BAD, and HRK), and it ultimately results in the release of cytochrome C and other pro-AP factors from the mitochondria into the cytoplasm, leading to the assembling of the apoptosome, which activates CAS9. Both apoptotic pathways converge in the execution pathway, where the activated effector caspases 3 and 7 are responsible for all cellular changes that define the AP phenomenon. Figure created only for this article by A.G.-M., with the BioRender.com tool, accessed on 10 October 2023. (BCL-2: B-cell lymphoma 2; TNF: tumor necrosis factor; Fas-L: Fas ligand; DISC: death-inducing signaling complex; CAS: caspase).

**Figure 3 jcm-13-00074-f003:**
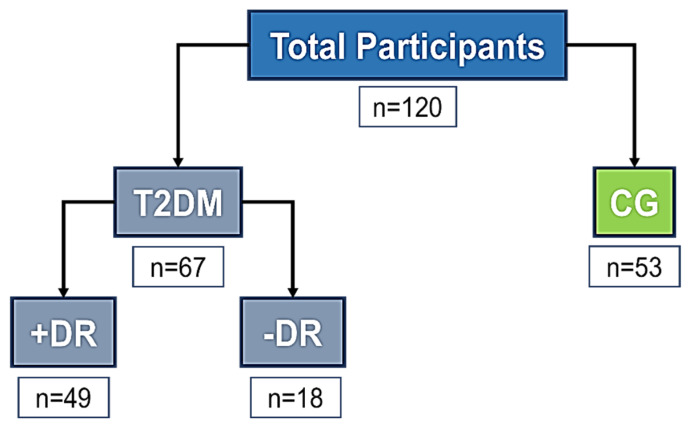
**Flowchart of the study participant groups and subgroups**.

**Figure 4 jcm-13-00074-f004:**
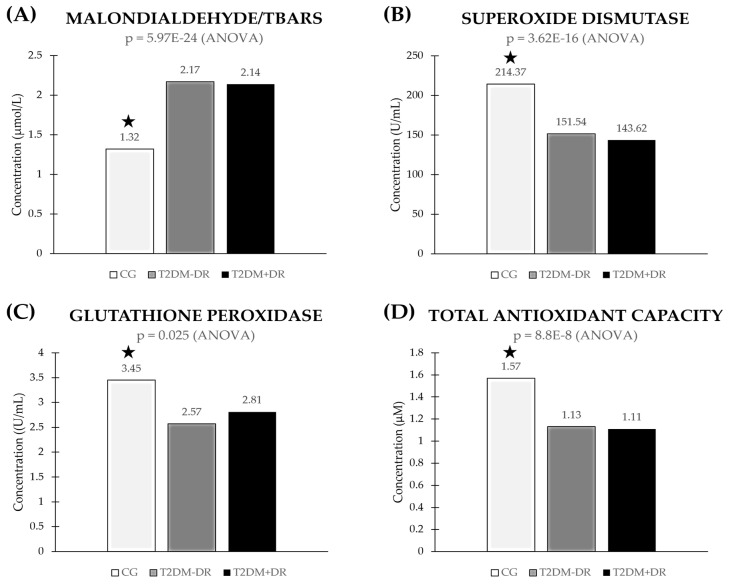
**Oxidative stress biomarkers in plasma samples of the study participants.** CG: control group; T2DM−DR: diabetics without retinopathy; T2DM+DR: diabetics with retinopathy; TBARS: thiobarbituric acid reactive substances; *p* < 0.02 for all parameters shown in this figure. * indicates statistically significant differences between groups or subgroups.

**Figure 5 jcm-13-00074-f005:**
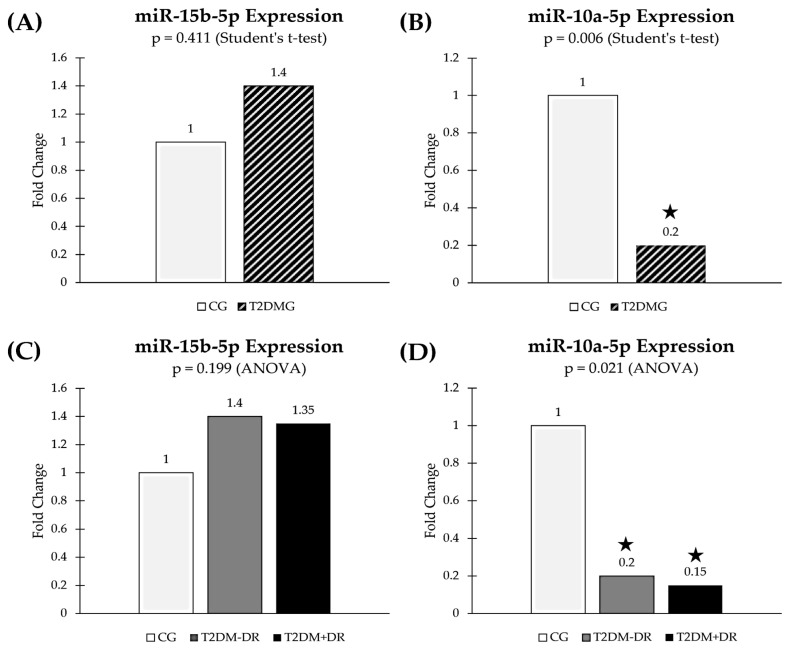
**Tear miRNA expression in the study participants**. (**A**,**B**) show the expression levels of miR-15b-5p and miR-10a-5p, respectively, in tears from the T2DM patients and the CG. (**C**,**D**) show the expression of miR-15b-5p and miR-10a-5p, respectively, between the T2DM+DR patients, the T2DM−DR patients, and the CG. miR: microRNA; CG: control group; T2DMG: type 2 diabetes group; T2DM−DR: diabetics without retinopathy; T2DM+DR: diabetics with retinopathy. * indicates statistically significant differences from the CG.

**Figure 6 jcm-13-00074-f006:**
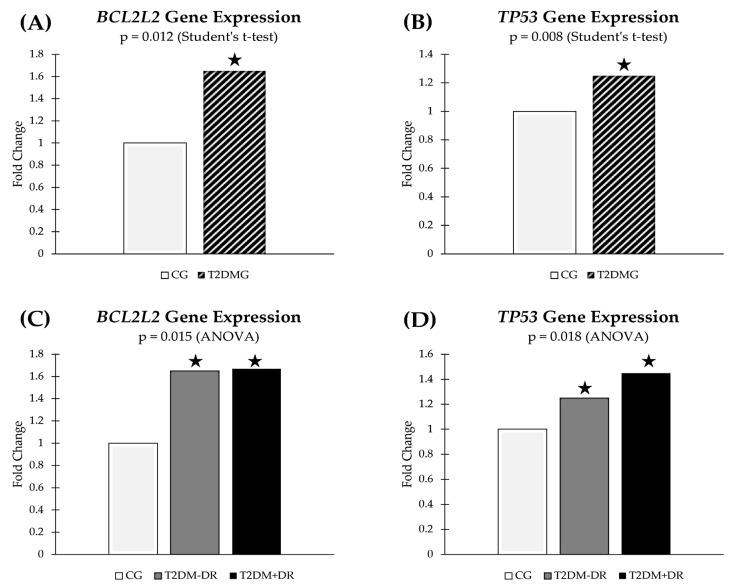
**Potential miRNA-regulated gene blood expression, as filtered by bioinformatic target prediction algorithms.** (**A**) Differential expression profile of *BCL2L2* gene between the TD2M group and the CG. (**B**) Differential expression profile of *TP53* gene between the TD2M group and the CG. (**C**) Differential expression profile of *BCL2L2* gene between the T2DM+DR patients, the T2DM−DR patients, and the CG. Pairwise comparisons: T2DM−DR vs. CG, *p* = 0.020; T2DM+DR vs. CG, *p* = 0.009; T2DM+Dr vs. T2DM−DR, *p* = 0.166. (**D**) Differential expression profile of *TP53* gene between the T2DM+DR patients, the T2DM−DR patients, and the CG. Pairwise comparisons: T2DM−DR vs. CG, *p* = 0.033; T2DM+DR vs. CG, *p* = 0.001; T2DM+Dr vs. T2DM−DR, *p* = 0.007. BCL2L2 gene: B-cell lymphoma 2L2 gene; TP53 gene: tumor protein 53 gene; T2DM−DR: diabetics without retinopathy; T2DM+DR: diabetics with retinopathy. * indicates statistically significant differences from the CG in each diagram. Connections indicate statistically significant differences between groups.

**Table 1 jcm-13-00074-t001:** Inclusion and exclusion criteria for the study participants.

INCLUSION	EXCLUSION
Individuals with T2DM aged 40–80 years, for the T2DM group. Individuals with T2DM and DR for the T2DM+DR subgroup. Individuals with T2DM without DR for the T2DM−DR subgroup. Healthy individuals for the CG. Those individuals with complete and precise data in their medical history.	Individuals with other DM types or who are <40 years or >80 years of age. Patients suffering other eye diseases than DR or other systemic disorders that do not fit with the study purpose. Patients under treatment that may interfere with the results of the study. Patients that have undergone laser and/or eye surgery in previous years. Individuals with incomplete and/or confusing clinical data, or for whom it is impossible to obtain a complete and thorough clinical history.

DM: diabetes mellitus; T2DM: type 2 diabetes mellitus; DR: diabetic retinopathy; T2DM+DR: type 2 diabetes mellitus group of participants having diabetic retinopathy; T2DM−DR: type 2 diabetes mellitus group of participants having diabetic retinopathy; CG: control group.

**Table 2 jcm-13-00074-t002:** Personal/familial characteristics and lifestyle in the study participants.

Variables	CG	T2DM−DR	T2DM+DR	*p*-Value (χ^2^)
HBP (%)	-	44 (62.9%)	44 (75.9%)	4.92 × 10^−18^ *
FH of T2DM (%)	14 (23.3%)	16 (22.9%)	28 (48.3%)	5.35 × 10^−19^ *
FH of HBP (%)	28 (46.7%)	16 (22.9%)	34 (58.6%)	1.63 × 10^−16^ *
Alcohol intake (%)	52 (86.6%)	42 (60%)	28 (31%)	2.21 × 10^−7^ *
Tobacco use (%)	30 (50%)	28 (40%)	10 (17.2%)	0.000055 *
BMI (kg/m^2^)	23.57 ± 3.28	26.72 ± 3.54	26.84 ± 2.80	1.5091 × 10^−8^ *
Physical exercise (%)	42 (70%)	32 (55.7%)	4 (6.9%)	2.91 × 10^−12^ *

CG: control group; T2DM−DR: diabetics without retinopathy; T2DM+DR: diabetics with retinopathy; HBP: high blood pressure; FH: familial history; BMI: body mass index; p-c2: *p*-value (chi square). * indicates statistically significant differences between groups or subgroups according to the chi-squared test.

**Table 3 jcm-13-00074-t003:** Ophthalmologic parameters of the study participants.

Variables	CG	T2DM−DR	T2DM+DR	*p*-Value (ANOVA)
BCVA (LogMAR)	0.037 ± 0.09	0.042 ± 0.08	0.127 ± 0.22	0.014 *
IOP (mmHg)	15.7 ± 2.4	15 ± 2	14.9 ± 1.9	0.912
CST (mm)	258.26 ± 31.23	252.51 ± 20.39	285.1 ± 45.17	3.86 × 10^−8^ *
CV (mm^3^)	0.205 ± 0.019	0.208 ± 0.06	0.235 ± 0.013	1.25 × 10^−28^ *

CG: control group; T2DM−DR: type 2 diabetics without retinopathy; T2DM+DR: diabetics with retinopathy; BCVA: best corrected visual acuity; IOP: intraocular pressure; CST: central subfield macular thickness; CV: central macular volume. * indicates statistically significant differences between groups or subgroups.

**Table 4 jcm-13-00074-t004:** Clinical biochemistry.

Variables	CG	T2DM−DR	T2DM+DR	*p*-Value
Glycemia (mg/dL)	91 ± 9	98 ± 14	97 ± 13	0.011 *
HbA1c (%)	5.64 ± 0.37	7.07 ± 0.41	7.25 ± 0.66	3.63 × 10^−46^ *
T Chol (mg/dL)	170 ± 40	159 ± 22	172 ± 22	0.029 *
VLDL (mg/dL)	18 ± 5	32 ± 5	42 ± 9	8.87 × 10^−49^ *
LDL (mg/dL)	95 ± 31	103 ± 23	112 ± 19	0.002 *
HDL (mg/dL)	59 ± 13	53 ± 8	55 ± 10	0.008 *
TRIG (mg/dL)	83 ± 30	91 ± 9	91 ± 13	0.042 *
Urea (mg/dL)	35 ± 8	38 ± 11	38 ± 11	0.391
CREAT (mg/dL)	0.93 ± 0.18	0.93 ± 0.21	0.94 ± 0.25	0.992
Iron (mg/dL)	76 ± 22	89 ± 31	77 ± 23	0.125

CG: control group; T2DM−DR: diabetics without retinopathy; T2DM+DR: diabetics with retinopathy; HbA1c: glycosylated hemoglobin; T Chol: total cholesterol; VLDL: very low-density lipoprotein; LDL: low-density lipoprotein; HDL: high-density lipoprotein; TRIG: triglycerides; CREAT: creatinine. * indicates statistically significant differences between groups or subgroups.

**Table 5 jcm-13-00074-t005:** Bivariate correlation analyses of the OCT examination and biochemical parameters from the study groups and subgroups.

	CST (mm)		CV (mm^3^)	
Biochemical Data	PCC	PPC *p*-Value	PCC	PPC *p*-Value
Basal glycemia	−0.026	0.726	0.050	0.493
HbA1c	0.144 *	0.049	0.275 **	<0.001
T Chol	0.110	0.132	0.262 **	<0.001
VLDL	0.277 **	<0.001	0.443 **	<0.001
LDL	0.189 **	0.009	0.352 **	<0.001
HDL	−0.044	0.552	−0.046	0.535
TRIG	0.094	0.197	0.085	0.248
BMI	−0.046	0.531	0.123	0.094

CST: central subfield thickness of the macula in OCT; CV: central volume of the macula in OCT; PCC: Pearson correlation coefficient; HbA1c: glycosylated hemoglobin; T Chol: total cholesterol; VLDL: very low-density lipoprotein; LDL: low-density lipoprotein; HDL: high-density lipoprotein; TRIG: triglycerides; BMI: body mass index. ** Correlation significance at the 0.01 level (2 tails). * Correlation significance at the 0.05 level (2 tails).

**Table 6 jcm-13-00074-t006:** Comparison of tear miRNA expression between groups and subgroups after sequencing.

T2DM vs. CG				T2DM+DR vs. T2DM−DR			
Upregulated	*p*-Value	Downregulated	*p*-Value	Upregulated	*p*-Value	Downregulated	*p*-Value
hsa-miR-155-5p	0.00050289	**hsa-miR-10a-5p**	0.00021554	hsa-miR-147b	0.00033305	hsa-miR-342-3p	0.0006826
hsa-miR-4488	0.00215255	hsa-miR-195-3p	0.00344617	hsa-miR-31-5p	0.00356039	hsa-miR-148a-3p	0.00137003
hsa-miR-4516	0.00267474	hsa-miR-135a-5p	0.01745868	hsa-miR-34a-5p	0.00550053	hsa-miR-27a-5p	0.00463293
hsa-miR-92b-5p	0.00352394	hsa-miR-320a	0.01795334	hsa-miR-4436b-3p	0.00667778	hsa-miR-423-5p	0.02206753
**hsa-miR-15b-5p**	0.01224091	hsa-miR-342-5p	0.01846176	hsa-miR-3158-3p	0.0076641	hsa-miR-9-3p	0.02287876
hsa-miR-139-5p	0.01696883	hsa-miR-486-5p	0.02715084	hsa-miR-508-3p	0.00797464	hsa-miR-195-3p	0.02605848
hsa-miR-203	0.04464176			hsa-miR-155-5p	0.01040889	hsa-miR-4794	0.02879332
hsa-miR-378a-3p	0.04503972			hsa-miR-450b-5p	0.01488504	hsa-miR-493-3p	0.02879332
**T2DM−DR vs. CG**				hsa-miR-20b-5p	0.01673733	hsa-miR-550a-3p	0.02879332
**Upregulated**	***p*-value**	**Downregulated**	***p*-value**	hsa-miR-211-5p	0.01899113	hsa-miR-204-3p	0.03643016
hsa-miR-155-5p	0.00048824	**hsa-miR-10a-5p**	0.00022013	hsa-miR-1287	0.02255178	hsa-miR-3648	0.03705494
**hsa-miR-15b-5p**	0.00405801	hsa-miR-452-5p	0.00069978	hsa-miR-203	0.02305019	hsa-miR-625-5p	0.03804594
hsa-miR-375	0.01056473	hsa-miR-186-5p	0.00282549	hsa-miR-504	0.02427954	hsa-miR-4638-3p	0.04095184
hsa-miR-708-3p	0.01085626	hsa-miR-34a-5p	0.01578009	hsa-miR-455-5p	0.02587983	hsa-miR-451a	0.04272808
hsa-miR-1260a	0.01168696	hsa-miR-324-3p	0.01956076	hsa-miR-505-3p	0.02702551		
hsa-miR-184	0.02463651	hsa-miR-195-3p	0.02066843	hsa-miR-30c-2-3p	0.02736074		
hsa-miR-92b-5p	0.03447955	hsa-miR-27a-5p	0.02106449	hsa-miR-550a-3-5p	0.0277247		
		hsa-miR-103a-3p	0.02243327	**hsa-miR-15b-5p**	0.03226176		
		hsa-miR-30e-5p	0.02258532	hsa-miR-651	0.03254246		
		hsa-miR-29b-2-5p	0.02387301	hsa-miR-720	0.03528564		
		hsa-miR-342-5p	0.02391494	hsa-miR-675-3p	0.03571653		
		hsa-miR-193b-5p	0.02683451	hsa-miR-4662a-5p	0.0365617		
**T2DM+DR vs. CG**				hsa-miR-942	0.03822863		
**Upregulated**	***p*-value**	**Downregulated**	***p*-value**	hsa-miR-330-5p	0.03836754		
**hsa-miR-15b-5p**	0.00038556	**hsa-miR-10a-5p**	0.00010335	hsa-miR-1278	0.03912887		
hsa-miR-155-5p	0.00041997	hsa-miR-195-3p	0.00107758	hsa-miR-30b-3p	0.04009894		
hsa-miR-342-3p	0.00233141	hsa-miR-451a	0.00651537	hsa-miR-4446-3p	0.04031481		
hsa-miR-27a-5p	0.00360745	hsa-miR-203	0.01323233	hsa-miR-19a-3p	0.04039565		
hsa-miR-423-3p	0.01721141	hsa-miR-211-5p	0.02409484	hsa-miR-130b-5p	0.04109527		
hsa-miR-328	0.0327679	hsa-let-7a-3p	0.02572669	hsa-miR-92b-5p	0.04321649		
		hsa-miR-375	0.04351622	hsa-miR-27b-5p	0.0448958		
		hsa-miR-184	0.04767438	hsa-miR-3126-5p	0.04586702		
		hsa-miR-204-3p	0.04781022	hsa-miR-501-3p	0.04778805		
		hsa-miR-324-3p	0.04801066				
		hsa-miR-708-3p	0.04923371				

This table includes only miRNAs with significant differential expressions between the study groups. T2DM: type 2 diabetes mellitus; +DR: with diabetic retinopathy; −DR: without diabetic retinopathy; CG: reference group.

## Data Availability

Data available upon reasonable request.

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
