# Peer review of "Oxidative Stress Mediates Epigenetic Modifications and the Expression of miRNAs and Genes Related to Apoptosis in Diabetic Retinopathy Patients"

_jcm, 2023, doi:10.3390/jcm13010074_

Round 1
Reviewer 1 Report
Comments and Suggestions for Authors
In my opinion manuscript jcm-2744860) Oxidative stress mediates epigenetic modifications and the expression of miRNAs and genes related to apoptosis in diabetic retinopathy patients is an interesting research paper on the important subject.
In this study compared demographics, clinical, biochemical, and molecular genetic data obtained from diabetic type 2 patients without diabetic retinopathy (T2DM DR), T2DM subjects with DR (T2DM+DR), and control individuals (CG).
The authors of the study, examining 120 Caucasian individuals (T2DM n=67 and control group n=53), analysed oxidative stress parameters such as SOD and GPx activity, serum level of TAS and MDA, caspase 3
level (CAS3) as biomarker of apoptosis, two miRNA s: miR 10a 5p and miR 15a 5b fingerprints, and their predicted target genes BCL2L2 and TP53.
This manuscript, in which the authors' goal is to try to find links between oxidative stress, apoptosis and epigenetic modifications in patients with diabetic retinopathy, is a timely piece of work with high clinical importance.
In summary , the article is important for understanding the pathomechanism of DR, but the need to increase the number of research groups should be emphasized.
The manuscript requires necessary editorial corrections.
Comments:
1.
Quality of all figures is low and needs improvement, especially the numerical data on the ordinate axis from Figure 4.
2.
table headings: 2, 3 and 4 (variables):
GC; T2DM-RD; T2DM+RD do not match table legends: CG; T2DM-DR; T2DM+DR
3.
according to the information in the abstract section, two miRNAs examined were miR10a-5p and miR15b-5p in the text, the authors mistake the names of analyzed miRNAs, e.g. line 443, 451, 453, 590, 597, 602 .....etc
Therefore, the Authors should take a position on this matter and make corrections.
4.
line 597 - there is information about miR-15a-5p (?)
is this the miR15b-5p being analyzed? if so, according to Fig. 5 (a) and (c) the level of this miRNA is up-regulated, not down-regulated (?)
5.
for Figure 5 (a) and (b), the description of the T2DMG parameter is missing in the legend
I understand this is a T2DM group (T2DMG)?
6.
line 576 please edit the correct entry regarding superoxide anion and hydroxyl radical (O2.-) and (OH) - this is not a correct description of these free radicals
Author Response
ANSWERS TO THE EDITOR AND THE REVIEWERS
The authors are extremely grateful to The Reviewers and The Editor of the Journal of Clinical Medicine, for the carved time out of their busy schedules and evaluating our work. Also, we want to thank The Reviewers and The Editor for the opportunity to clarify our research and to improve our manuscript, by offering constructive criticism and detailing our article limitations. We have addressed each, and every concern The Reviewers have raised, that have been fully taken into consideration. The optimized revised version of our work is now sent to the Journal of Clinical Medicine, for evaluation.
We have responded point by point to the comments and suggestions. We have also reviewed the English language through the ms., and illustrations, as suggested. The corresponding answers of the authors to The Reviewers are listed below.
REFEREE 1.
The authors are very grateful to The Reviewer, for the time invested in this review and for the comments you gave us to improve our manuscript, that has been rewritten. We have introduced several new sentences and two new references (marked), that may help clarifying important points in current version of our work. Thank you so much for outstanding help.
The Referee 1 wrote:
In my opinion manuscript jcm-2744860) Oxidative stress mediates epigenetic modifications and the expression of miRNAs and genes related to apoptosis in diabetic retinopathy patients is an interesting research paper on the important subject. In this study compared demographics, clinical, biochemical, and molecular genetic data obtained from diabetic type 2 patients without diabetic retinopathy (T2DM DR), T2DM subjects with DR (T2DM+DR), and control individuals (CG). The authors of the study, examining 120 Caucasian individuals (T2DM n=67 and control group n=53), analysed oxidative stress parameters such as SOD and GPx activity, serum level of TAS and MDA, caspase 3 level (CAS3) as biomarker of apoptosis, two miRNA s: miR 10a 5p and miR 15a 5b fingerprints, and their predicted target genes BCL2L2 and TP53. This manuscript, in which the authors' goal is to try to find links between oxidative stress, apoptosis and epigenetic modifications in patients with diabetic retinopathy, is a timely piece of work with high clinical importance.
In summary, the article is important for understanding the pathomechanism of DR, but the need to increase the number of research groups should be emphasized.
The manuscript requires necessary editorial corrections.
Comments:
- Quality of all figures is low and needs improvement, especially the numerical data on the ordinate axis from Figure 4.
We appreciate your feedback and your advice. It is true: the quality of the figures NEEDS to be improved, and we have worked in that direction. In addition, we have added some missing words in the figure legends (marked). Please, see the new results section (pages: 9-15), Thank you again for your help.
- Table headings: 2, 3 and 4 (variables): GC; T2DM-RD; T2DM+RD do not match table legends: CG; T2DM-DR; T2DM+DR
Regarding this point, we agree with your important suggestion, and the table headings has been now changed (marked), as well as other misspellings that we detected when reviewing the ms. Thanks for your help.
- According to the information in the abstract section, two miRNAs examined were miR10a-5p and miR15b-5p in the text, the authors mistake the names of analyzed miRNAs, e.g. line 443, 451, 453, 590, 597, 602 .....etc. Therefore, the Authors should take a position on this matter and make corrections.
We thank The Reviewer for calling our attention to this relevant point. Sorry, it was a mistake. The miR10a-5p and the miR15b-5p are the ones that we have studied in this population. This important failure has been corrected in the abstract, and through the text and figures.
- Line 597 - there is information about miR-15a-5p (?) is this the miR15b-5p being analyzed? if so, according to Fig. 5 (a) and (c) the level of this miRNA is up-regulated, not down-regulated (?)
Thanking The Reviewer for this important comment, as in our previous response, we have corrected this important matter in the new version of the ms. Only both, the miR10a-5p and miR15b-5p have been studied herein. Many thanks for your positive criticism to our work.
- For Figure 5 (a) and (b), the description of the T2DMG parameter is missing in the legend I understand this is a T2DM group (T2DMG)?
We apologize for this fault. The corresponding legend to T2DMG has been incorporated, according to The Reviewer comments and all descriptions of the parameters have been reviewed throughout the text and illustrations (marked).
- Line 576 please edit the correct entry regarding superoxide anion and hydroxyl radical (O2.-) and (OH) - this is not a correct description of these free radicals.
We apologize, again, for this fault. It was a typographical error (former page 576), but we now have reviewed and corrected any type of entry regarding these two free radicals, as well as we did it for the corresponding position of the symbols, superscripts, and subscripts: (O2•−) and (·OH) on the new lines 592-593 (marked).
Thank you so much for raising the above questions. We sincerely hope that in the present revised form our work can be ready for being accepted for publication.
Reviewer 2 Report
Comments and Suggestions for Authors
The article by Karam-Palos et al aims to study the early diabetic retinopathy markers in the Tear samples from DR patients, and correlate oxidative stress and apoptosis processes to the onset of DR. Though the study reports interesting miRNA markers for non-invasive diagnosis of DR, it needs robust validation of the data. All gene expression data in the manuscript should be carefully analyzed.
As tear is produced from the anterior segment of the eye, the actual contact between tear and retina is not very significant although recent studies are showing some changes in the tear proteomics under DR condition but the miRNA profiles are not studied in detail yet. The authors could have included a patient group from proliferative diabetic retinopathy who would be undergoing vitrectomy (along with a disease matched control like diabetic macular hole) and the vitreous could have been used to validate their findings on Tear sample. This would have give an idea whether the expression of Tear identified miRNAs are similarly altered in the vitreous as well, since the authors hypothesize that miR-15-a could be profibrotic and lead to proliferative DR.
The study did not show any data related to epigenetic modifications, therefore the current title of the manuscript is not appropriate. It should be modified to reflect the findings presented in the report.
In the abstract, line 45, please remove 'to identify potential epigenetic modification in the T2DM eyes' as there were no data presented to this claim. Also, please mention in the abstract that the miRNAs were identified from Tear samples.
Fig.4 image quality is poor. The units and values are not legible. It would be better to label each graph individually (Fig 4 a,b,c and d) and write the legend accordingly.
It is important to include the RNAseq data obtained from the tear samples and show how significant was the two selected miRNAs.
Fig.5 - c and d graphs show no significant differences between T2DM-DR and T2DM+DR. The presented data implies that the miRNAs are differentially expressed between the control and diabetic condition. However, there is very marginal downregulation of both miRs in the DR condition. Please provide each datapoint as a dotplot (for c and d) to show the actual differences. If there is no significant changes in the miRNA expression between T2DM-DR and T2DM+DR, then this part of work has no importance. Line 451, 453 and Fig.4a, the miR-15a-5p name is written in 3 different ways. Please clarify whether it is 15a or 15b.
Fig.6 B and D - again the difference between T2DM-DR and T2DM+DR is not significant. Please replot the graph and do the statistical analysis properly.
References are not formatted well. There are 80 refs in the text and the list shows 81. Check Line 899, and other parts of reference section.
The readability of the manuscript will improve if the writing is concise and not over extrapolating the results to the disease progression and management as this work is very preliminary with limited sample size.
Comments on the Quality of English Language
There are a few grammatical and syntax errors in the manuscript.
Author Response
ANSWERS TO THE EDITOR AND THE REVIEWERS
The authors are extremely grateful to The Reviewers and The Editor of the Journal of Clinical Medicine, for the carved time out of their busy schedules and evaluating our work. Also, we want to thank The Reviewers and The Editor for the opportunity to clarify our research and to improve our manuscript, by offering constructive criticism and detailing our article limitations. We have addressed each, and every concern The Reviewers have raised, that have been fully taken into consideration. The optimized revised version of our work is now sent to the Journal of Clinical Medicine, for evaluation.
We have responded point by point to the comments and suggestions. We have also reviewed the English language through the ms., and illustrations, as suggested. The corresponding answers of the authors to The Reviewers are listed below.
REFEREE 2.-
The article by Karam-Palos et al aims to study the early diabetic retinopathy markers in the Tear samples from DR patients, and correlate oxidative stress and apoptosis processes to the onset of DR. Though the study reports interesting miRNA markers for non-invasive diagnosis of DR, it needs robust validation of the data. All gene expression data in the manuscript should be carefully analyzed.
The Referee 2 wrote:
1.1.- As tear is produced from the anterior segment of the eye, the actual contact between tear and retina is not very significant although recent studies are showing some changes in the tear proteomics under DR condition, but the miRNA profiles are not studied in detail yet.
We appreciate the feedback and the advice of The Reviewer. This point is pivotal to us, and a controversial topic (see below some references) for the real value of the biological samples from outside/inside the eyes, as well as for the penetration power of some drugs from the anterior to the posterior eye segments as worldwide known. Thank you so much for calling our attention to this issue.
- Hagan S, Martin E, Enríquez-de-Salamanca A. Tear fluid biomarkers in ocular and systemic disease: potential use for predictive, preventive, and personalised medicine. EPMA J. 2016;7(1):15.
- Löscher M, Seiz C, Hurst J, Schnichels S. Topical Drug Delivery to the Posterior Segment of the Eye. Pharmaceutics. 2022;14(1):134
- Wang L, Zhou MB, Zhang H. The Emerging Role of Topical Ocular Drugs to Target the Posterior Eye. Ophthalmol Ther. 2021;10(3):465-494
- von Thun und Hohenstein-Blaul N, Funke S, Grus FH. Tears as a source of biomarkers for ocular and systemic diseases. Exp Eye Res, 2013; 117:126-137.
Our general response to this question is given below:
The minimal invasiveness characteristic, relatively easy collection and storing procedures of the tear film samples, have spread their use for the diagnosis/therapy of eye diseases, far than their role in the ocular surface disorders. Meanwhile, many researchers have identified different types of molecules that have characterized as disease specific biomarkers with diagnostic potential, on the basis that ocular-systemic diseases can change the proteome patterns of tear fluid. Among the above diseases: thyroid orbitopathy, aniridia, glaucoma and other neurodegenerative pathologies (Alzheimer, Parkinson, multiple sclerosis), as well as systemic disorders (cancer, cystic fibrosis, scleroderma, diabetes, etc). We agree with this hypothesis, and, in our opinion, pathological changes occurring in the choroid, retina and vitreous can be reflected in the whole eye, including the ocular surface components, either by diffusion through the sclera or cornea [Ambati J, Canakis CS, Miller JW, Gragoudas ES, Edwards A, Weissgold DJ, Kim I, Delori FC, Adamis AP. Diffusion of high molecular weight compounds through sclera. Invest Ophthalmol Vis Sci. 2000;41(5):1181-5], or via the local and systemic circulation, and vice versa. In an intend to clarify this point to the readers, a new sentence has been enclosed on the new version of our ms. (page 18).
Since 2013, we have published 12 articles regarding the tear film samples for the diagnosis of ocular diseases. Our multidisciplinary group is really interested in evaluating molecules and genes involved in oxidative stress, inflammation, angiogenesis, and apoptosis, and its implications in diabetic retinopathy and diabetic macular edema. We have just published an article [Andrés-Blasco I, Gallego-Martínez A, Machado X, Cruz-Espinosa J, Di Lauro S, Casaroli-Marano R, Alegre-Ituarte V, Arévalo JF, Pinazo-Durán MD. Oxidative Stress, Inflammatory, Angiogenic, and Apoptotic molecules in Proliferative Diabetic Retinopathy and Diabetic Macular Edema Patients. Int J Mol Sci. 2023 May 4;24(9):8227], in which we investigated the correlation between the above players in the plasma and vitreous body from type 2 diabetics (T2DM) with proliferative diabetic retinopathy/diabetic macular edema, (PDRG/DMEG; n = 112), and non-T2DM subjects as the surrogate controls (SCG; n = 48) that were programmed for vitrectomy, either by having PDR/DME or macular hole (MH)/epiretinal membrane (ERM)/rhegmatogenous retinal detachment. This was done, and the integrated ophthalmologic (ocular examination and fundus imaging) and biochemical-molecular plasma ad vitreous samples data showed that chronic hyperglycemia induces anomalies in a wide spectrum of biochemical pathways, the majority of these with cross-talk between them. We demonstrated that the MDA, 4HNE, VEGF, and IL6 concentrations increased whilst the superoxide dismutase, catalase, and TAC decreased in the plasma and vitreous samples from PDR and DME patients versus the surrogate controls, with a clear correlation of the increased pro-oxidants, proangiogenic and proinflammatory markers and decreased antioxidants between the plasma and vitreous body of diabetics versus the surrogate controls. Finally, we concluded that the above molecules are biomarker candidates for distinguishing T2DM patients at risk of PDR, DME and vision loss, by using either plasma or vitreous body samples.
Furthermore, in our most recent research (Del Valle J, Andrés-Blasco I, Gallego-Martínez A, Di Lauro S, Casaroli-Marano S, Pinazo-Durán MD. New insights in Endothelial Dysfunction in Diabetic Retinopathy and Diabetic Macular Edema. Ophthalmic Res 2023, 66(suppl 1):1–72) we have also seen that pro-inflammatory molecules (IL-6, tumor necrosis factor a) showed a differential expression profile in T2DM + Non PDR versus T2DM -Non PDR and the comparison group in both the tear film and vitreous body samples. In fact, we demonstrated that the pathogenic mechanisms involving inflammation/angiogenesis/apoptosis, regulated by the hsa-miR-155-5p, hsa-miR-15b-5p and hsa-miR-10a-5p, and its respective target genes SOCS6, VEGF-A, and BCL2L2, expressed in tears, plays important roles in structural and functional changes associated with microvascular endothelial dysfunction in DR/DME. We may suggest the above players as presumptive diagnostic-prognostic biomarkers for the diabetic eyes.
Kumar et al., [Kumar S, Joshi MB, Kaur I. Protocol and Methods Applicable to Retinal Vascular Diseases. Methods Mol Biol. 2023;2625:71-78], raised the unavailability of suitable biological retinal tissue need for research, for obvious ethical issues. The authors explored vitreous humor and tear film for studying lipidomic alterations in different ocular diseases, by using tandem mass spectrometry approaches. They focused on a simplified protocol for extracting sufficient lipids/metabolites from vitreous humor and tear samples obtained from patients and their subsequent mass spectrometry analysis, validating the usefulness of these two biological samples for identifying lipidomic alterations in ocular pathologies. Touching with this same topic on the conflicting reports regarding the relationship between tear and plasma urea levels, an interesting article from Germany, India, and Switzerland [Singh S, Hammer CM, Paulsen F. Urea and ocular surface: Synthesis, secretion and its role in tear film homeostasis. Ocul Surf. 2023;27:41-47] reported that urea can be detected in the eye fluids (tears, aqueous humor, and vitreous). The authors specify that most of the urea in the aqueous humor and vitreous is considered an ultrafiltrate from the blood, but the presence of urea transporters and urea-synthesizing enzymes in the lacrimal gland, meibomian glands, conjunctiva, and cornea suggests ureagenesis occurring at the same ocular surface. Furthermore, Singh et al., stated that the urea concentration on the ocular surface is influenced by blood urea concentration, the amount of urea released by the tear fluid, tear evaporation, and arginase concentration in the tears.
Nevertheless, further research is needed to clarify this important “hot topic” in ophthalmology, that is currently under controversy worldwide.
1.2.- The authors could have included a patient group from proliferative diabetic retinopathy who would be undergoing vitrectomy (along with a disease matched control like diabetic macular hole) and the vitreous could have been used to validate their findings on Tear sample. This would have given an idea whether the expression of Tear identified miRNAs are similarly altered in the vitreous as well, since the authors hypothesize that miR-15-a could be profibrotic and lead to proliferative DR.
Thank you so much to The Reviewer for this important comment. In fact, in our recent work [Andrés-Blasco et al., Int J Mol Sci. 2023] we did it, by investigating the correlation of molecules/genes in plasma and vitreous body from type 2 diabetics (T2DM) with PDR/DME (n=112), and non-T2DM subjects as the surrogate controls (n=48) that were programmed for vitrectomy, either by having PDR/DME or macular hole (MH)/epiretinal membrane (ERM)/rhegmatogenous retinal detachment. We also did it in our most recent publication [Del Valle et al., Ophthalmic Res 2023], by studying the expression profile in T2DM+NPDR versus T2DM-NPDR and the comparison group in both the tear film and vitreous body samples. We are almost ready to send a new article in this sense. Thank you for outstanding positive criticism, comments, and suggestions.
- The study did not show any data related to epigenetic modifications, therefore the current title of the manuscript is not appropriate. It should be modified to reflect the findings presented in the report.
We thank the Reviewer for this suggestion. However, we would like respectfully to say the Reviewer that there are major epigenetic mechanisms of regulation, and one of these (probably the least known) is mediated by small non-coding RNAs, such as the miRNAs. In this work we have analyzed the gene expression (in blood samples) and the miRNA expression profile (in tear samples) in T2DM patients, in comparison with the gene expression and the miRNA expression profile in healthy controls. Therefore, the differences observed in gene expression can be understood as epigenetic modification regulated by a different miRNA signature, as shown in the results section. Some sentences have been included throughout the ms and illustrations to clarify this important issue (marked in blue). We sincerely thank the Reviewer for having pointed out this weakness of our text and giving us the possibility of improving it so that it is more understandable for readers.
3- In the abstract, line 45, please remove 'to identify potential epigenetic modification in the T2DM eyes' as there were no data presented to this claim.
We thank again the Reviewer for calling our attention again to this important point, and the possibility of trying to improve our work, so that in this new version this issue may be better followed by the readers. However, we would respectfully ask the Reviewer to maintain our title, and the results obtained from the miRNAs and genes in relation to this topic. As we said in the previous question, the differences in gene expression between the diabetic patients and the control subjects can be understood because of epigenetic modification (mostly induced by oxidative stress) regulated by the action of miRNAs. We have added some sentences regarding this important topic through the text and figures (marked in blue).
Also, please mention in the abstract that the miRNAs were identified from Tear samples.
Thanking The Reviewer for outstanding care, we did it, and marked in blue.
- Fig.4 image quality is poor. The units and values are not legible. It would be better to label each graph individually (Fig 4 a,b,c and d) and write the legend accordingly.
Thank you so much to the Reviewer for calling our attention to this point. The figure 4 has been newly done, according to the Reviewer´s comments.
5.1. Line 451, 453 and Fig.4a, the miR-15a-5p name is written in 3 different ways. Please clarify whether it is 15a or 15b.
We agree with the Reviewer. We have done an extensive review of the text and illustrations. We failed in the miR-15 name. We have changed this, and the miR-1b-5p is now correctly named. Thank you so much for outstanding help.
5.2. It is important to include the RNAseq data obtained from the tear samples and show how significant was the two selected miRNAs.
The authors thank the Reviewer for this relevant question. In fact, we reviewed our data and we have added a new table (current table 5) with the sequencing data, in the results section (page 13). In this table we show the up- and down-regulated miRNAs that we have identified in tears from the study participants, and the groups of interest compared to the reference groups. Thus, this new table 5 has been described in the corresponding results section as follows: This table shows the miRNAs significantly upregulated or downregulated among the different study groups, based on the RNAseq results. The groups with superscript (1) are those used as the reference group in each comparison. The two miRNAs analyzed in this study (miR-10a-5p and miR-15b-5p) are highlighted in black. We selected the miRNA-10a-5p and miR-15b-5p for this study because they were the miRNAs downregulated and upregulated (respectively) in T2DM+DR with the highest statistical significance.
- Fig.5 - c and d graphs show no significant differences between T2DM-DR and T2DM+DR. The presented data implies that the miRNAs are differentially expressed between the control and diabetic condition. However, there is very marginal downregulation of both miRs in the DR condition. Please provide each datapoint as a dotplot (for c and d) to show the actual differences. If there is no significant changes in the miRNA expression between T2DM-DR and T2DM+DR, then this part of work has no importance.
We respectfully ask the Reviewer for maintaining this part of our work, as we consider that the findings are important and innovative, in the context of DM, T2DM and DR. As for the Reviewer comments, we have reviewed our data, and we have improved this part, in order to be better explained to the readers, emphasizing the noticeable or significant differential expression profile of the miRNAs between groups. In fact, after qRT-PCR analysis we observed a noticeable up-regulation of the miR-15b-5p in both diabetic subgroups (T2DM-DR and T2DM+DR) when compared to the CG (p=0.199, Kruskal Wallis test). Furthermore, we observed a slight decreasing trend of this miRNA tear expression in T2DM+DR than in the T2DM-DR (p=0.401, pairwise comparison), as shown in the figure 5C and conveniently explained in the legend. Also, after qRT-PCR analysis, we identified statistically significant differences (p=0.021) between the groups and subgroups (T2DM-DR vs T2DM+DR vs CG; Kruskal Wallis test) of the tear miR-10a-5p expression levels. After pairwise comparisons, a statistically significant down-regulation of this miRNA was observed in the T2DM-DR patients comparing with the CG (p=0.019). Likewise, a statistically significant down-regulation was observed in the T2DM+DR patients when compared with CG (p=0.009). The expression of this miRNA was notably reduced in T2DM+DR compared to T2DM-DR, although the difference did not reach statistical signification (p=0.227). But this finding is also important in the context of T2DM patients. Thanking the Reviewer for this pivotal point to consider, we agree with the Reviewer, and we are now designing a new work with a higher sample size, to address the validation of the preliminary results obtained in the present work.
- Fig.6 B and D - again, the difference between T2DM-DR and T2DM+DR is not significant. Please replot the graph and do the statistical analysis properly.
We thank the reviewer for pointing us of this error. The difference in the expression of the TP53 and BCL2L2 genes between the T2DM-DR vs T2DM+DR groups is statistically significant. The figure 6 only showed the p-value obtained in the Kruskal Wallis test. As a result of this review process, and the comments of the Reviewer, we have done an extensive editing of all data, and those not previously reflected have been incorporated to the new version of our text and illustrations. Thanks to The Reviewer for permitting us to modify the figure 6B,D, indicating the significant differences through a connection between groups. Likewise, we have included the p-values of the pairwise comparisons in the figure legend.
- References are not formatted well. There are 80 refs in the text and the list shows 81. Check Line 899, and other parts of reference section.
Thank you for this suggestion. We have reviewed the complete text and references and we have corrected the mistakes. Now this section has also been improved with two more precise references.
The readability of the manuscript will improve if the writing is concise and not over extrapolating the results to the disease progression and management as this work is very preliminary with limited sample size.
We agree with the Reviewer in this point, and we tried to do a more precise sentences through the text and illustrations, due to the limited sample size. Even though a small number of subjects were used for the study, markers with high diagnostic value could be screened. but we respectfully ask the Reviewer for maintaining the results of this work, because we focus, in part, on issues that have not been dealt with before, and we finally got some interesting data than have to be improved and validated in our next work with a higher sample size.
The abstract, introduction, results, discussion and material and methods sections have been rewritten and all the adjustments have been marked in blue in the corresponding set. Some new references have been also enclosed. We sincerely hope that in the present revised form our work can be ready for being accepted for publication. Thank you so much for raising the above questions. Thank you so much for positive criticism, suggestions, and comments from which we have learned a lot and thus we have improved our article.
Round 2
Reviewer 2 Report
Comments and Suggestions for Authors
The revised manuscript has addressed many of the queries except a few. The major concern is the choice of miRNA candidates that they have selected. As the theme of the paper is about DR, I wonder why didn't the authors choose the upregulated and downregulated miRs from T2DM+DR vs T2DM-DR1 group over T2DM+DR vs CG1 which includes the changes caused by T2DM as well. As per the data shown by the authors, miR10a and 15b in the tear samples are actually indicators of DM rather than DR. In the qPCR, those miRs are not showing much difference in the T2DM+DR vs T2DM-DR1. Despite any justification, readers will ponder this.
Comments on the Quality of English Language
Very few spelling and syntax errors are there.
Author Response
ANSWERS TO THE EDITOR AND THE REVIEWERS
Once again, we the authors would like to express our gratitude to The Reviewers and The Editor of the Journal of Clinical Medicine, for outstanding help to improve our work.
We have answered to the point raised by the Reviewer 2.
Our ms has been rewritten, and the English language has been revised. The new version is ready to be evaluated for publication.
REFEREE 2.-
The revised manuscript has addressed many of the queries except a few. The major concern is the choice of miRNA candidates that they have selected. As the theme of the paper is about DR, I wonder why didn't the authors choose the upregulated and downregulated miRs from T2DM+DR vs T2DM-DR1 group over T2DM+DR vs CG1 which includes the changes caused by T2DM as well. As per the data shown by the authors, miR10a and 15b in the tear samples are actually indicators of DM rather than DR. In the qPCR, those miRs are not showing much difference in the T2DM+DR vs T2DM-DR1. Despite any justification, readers will ponder this.
Thank you so much to The Reviewer for this important comment, again.
We chose the miR-15b-5p, involved in apoptosis-related changes of neurodegenerative disorders, because the NGS results and bioinformatics analysis confirmed that it was significantly upregulated in tears, in all comparisons between the study groups (please, see the new table 5).
The miR-10a-5p, a cell cycle and apoptosis regulator, was also chosen because the NGS results and bioinformatics analysis showed that it was significantly downregulated in the comparisons, except when comparing tears from diabetics with and without retinopathy, but in this case, a noticeable lower expression levels were observed in diabetics with retinopathy.
The pathogenic mechanisms of diabetic retinopathy are complex, but it is clearly stated that the microangiopathy is the result of adverse metabolic effects of sustained hyperglycemia. Therefore, diabetics are at risk of suffering retinopathy and vision loss. miRNAs are long-distance communicators, reprogramming agents, and pivotal regulatory molecules for cellular activities. It has been reported a wide range of circulatory miRNAs that have been found to be differentially expressed in blood as well as in body fluids in diabetics with and without retinopathy. Also, some miRNAs have been associated with DR severity. As a result of our research, we found that both miRNAs are altered in diabetics suffering OS (please, see figures 4 and 5) and their predicted genes. Some new sentences have been enclosed in the new version of the ms (lines 602-622).
Also, the material and methods section and subsections have been re-structured. Two new references have been also enclosed.
We sincerely hope that in the present revised form our work can be ready for being accepted for publication.